# A Multimarker Model for Aberrant Cardiac Geometry after Preeclampsia

**DOI:** 10.3390/jcm11071900

**Published:** 2022-03-29

**Authors:** Zenab Mohseni-Alsalhi, Sophie A. J. S. Laven, Emma B. N. J. Janssen, Anique L. Wagenaar, Sander M. J. van Kuijk, Marc E. A. Spaanderman, Chahinda Ghossein-Doha

**Affiliations:** 1Department of Obstetrics and Gynecology, School for Oncology and Developmental Biology, GROW, Maastricht University Medical Center (MUMC+), MD 6200 Maastricht, The Netherlands; sajs.laven@student.maastrichtuniversity.nl (S.A.J.S.L.); emma.janssen@mumc.nl (E.B.N.J.J.); anique.wagenaar@maastrichtuniversity.nl (A.L.W.); marc.spaanderman@mumc.nl (M.E.A.S.); c.ghossein@maastrichtuniversity.nl (C.G.-D.); 2Department of Clinical Epidemiology and Medical Technology Assessment, Maastricht University Medical Center (MUMC+), MD 6200 Maastricht, The Netherlands; sander.van.kuijk@mumc.nl; 3Department of Obstetrics and Gynecology, Radboud University Medical Center, MD 6200 Maastricht, The Netherlands; 4Department of Cardiology, Cardiovascular Research Institute Maastricht (CARIM), Maastricht University Medical Center (MUMC+), MD 6200 Maastricht, The Netherlands

**Keywords:** preeclampsia, sex differences, heart failure, cardiac (diastolic) dysfunction, microvascular dysfunction, inflammation, fibrosis

## Abstract

One out of four women with a history of preeclampsia shows abnormal cardiac remodeling consistent with subclinical heart failure (HF) in the first decade postpartum. Since these women are susceptible for developing remote symptomatic HF, development of a model for aberrant cardiac geometry as a first screening tool after delivery, is urgently needed. In this cross-sectional study, 752 preeclamptic women were included. Cardiovascular evaluation was conducted between six months and five years postpartum including cardiac ultrasound, systolic and diastolic blood pressure (SBP and DBP), plasma volume (PV) and biomarker assessment. We developed a multimarker model using uni- and multivariable linear regression and used the regression coefficients (RC) to develop a formula and estimate the aberrant cardiac remodeling in our population. Both SBP and PV were shown to be independently correlated with relative wall thickness (RWT) and left ventricular mass index (LVMi). C-reactive protein (CRP) and uric acid were independently correlated with RWT. Fibrinogen did not relate to either LVMi or RWT. This study displays markers of abnormal cardiac remodeling in former preeclamptic women, suggesting a combination of mechanical and biochemical factors that should be involved in worrisome chamber remodeling before clinical symptoms arise.

## 1. Introduction

Heart failure (HF), arrhythmias, and ischemia are progressive and lethal diseases that are more prevalent in individuals with hypertensive heart disease compared to the general population, but also more prevalent in women with a history of preeclampsia (PE) [1,2,3].

PE is thought to be an endothelial derangement superimposed upon pre-existing circulatory, metabolic, hemostatic and immunological abnormalities that often coincide with defective placentation and complicates 2–8% of all pregnancies [1]. These pre-existing factors are pathogenic features which are also known for their involvement in cardiovascular disease (CVD) [4,5]. Half of preeclamptic women have asymptomatic structural cardiac alterations or dysfunction within one to two years after pregnancy amongst concentric remodeling, left ventricular hypertrophy and diastolic dysfunction, compared to 8% of the general female population [6,7,8]. Although concentric remodeling is a compensatory mechanism in order to decrease end-systolic stress in response to an increased pressure load [9,10], it is accompanied by detrimental mechanisms such as connective tissue accretion, consisting of collagen and fibroblast deposition, which is not always reversible [11]. In fact, cardiac fibrosis can lead to both contractile dysfunction and eventually dysrhythmia and HF.

However, not all former preeclamptic women will develop CVD later in life, which makes current screening strategies, involving state of the art vascular and cardiac ultrasound to detect occult CV abnormalities an important step in clinical practice. Therefore, we developed a multimarker model to assess the association between aberrant cardiac geometry after PE. 

## 2. Materials and Methods

This cross-sectional study is conducted in women after a pregnancy complicated by PE between 1996 and 2014. The study protocol was approved by the medical ethical committee board of Maastricht University Medical Center+ (MUMC+) (MEC 0-4-049). At the MUMC+ informed consent related to the use of clinically obtained data for scientific analysis was given. In accordance with institutional guidelines, the procedures were followed, and we adhered to the principles of the Declaration of Helsinki and Title 45, US Code of Federal Regulations, Part 46, Protection of Human Subjects, Revised 13 November 2001, effective 13 December 2001.

We included women with PE defined as new onset hypertension with a systolic blood pressure (BP) ≥ 140 and/or diastolic BP ≥ 90 mm Hg and the presence of proteinuria exceeding 0.3 g/day occurring after 20 weeks of gestation in previously normotensive women [12]. Early onset PE was defined as diagnosing PE before 34 weeks of gestation. Late onset PE was defined as diagnosing PE at or after 34 weeks of gestation. A small for gestational age (SGA) infant was defined as a fetal birth weight at or below the 10th percentile according to the Dutch reference standard. Preterm birth was defined as a delivery before 37 weeks. Women were recruited at the regular six weeks postpartum follow-up in a tertiary referral hospital (MUMC+) in Maastricht, the Netherlands, and invited to participate in a cardiovascular evaluation program at least six months postpartum.

Patients were scheduled for screening after discontinuation of the use of oral contraceptives, and after stopping breastfeeding for at least four weeks or more. Women with a postpartum interval between six months to five years after the preeclamptic pregnancy and a minimum age of 18 years were included in the study. Exclusion criteria were chronic hypertension, auto-immune disease, or kidney disease prior to the first pregnancy. Women who suffered from hemolysis, elevated liver enzymes, and low platelets (HELLP) syndrome without PE, and intrauterine fetal death (IUFD) were also excluded from the study. 

### 2.1. Measurements

The postpartum assessment was performed in one session that started at 8.00 a.m. Women were instructed to refrain from drinking and eating at least 10 h prior to measurements. Clinical data on obstetric variables of the index pregnancy were collected from medical files or self-report. Additional obstetrical history, medical history, lifestyle, and use of medication were retrieved by self-report. Height, weight, and blood pressure were measured by an experienced nurse. Body Mass Index (BMI) was calculated by dividing weight (kg) by squared height (meters). After five minutes of rest, arterial blood pressure was measured in sitting position on the right arm by a semiautomatic oscillometric device (Dinamap Vital Signs Monitor 1846; Critikon, Tampa, FL, USA). The median value of 11 measurements in a period of 30 min was used. When the arm circumference ranged between 27.5 and 36.5 cm a cuff size of 13.5 × 30.7 cm was used and when the arm circumference ranged between 35.5 and 46 cm a cuff size of 17 × 38.6 cm was used. 

Patient baseline and obstetric characteristics were registered during the screening. Hypertension was defined as a systolic blood pressure ≥ 140 and/or diastolic blood pressure ≥ 90 mm Hg or the use of antihypertensive drugs. Prehypertension was defined as SBP of 120–139 mm Hg and/or DBP of 80–90 mm Hg. Plasma volume (PV) was measured with the I-125 human serum albumin method (HSA) hemodilution technique and indexed for body surface area (BSA) [9]. Obesity was defined as a BMI of ≥30 kg/m^2^. Total fat mass was estimated by multiplying the height in meters with ((1.2 × BMI) + (0.23 × age) − 5.4). The following formula was used to calculate BSA: BSA (m^2^) = 0.007184 × height (cm)^0.725^ × height (kg)^0.425^.

Urine was collected in the 24 h preceding the measurements. The 24-h urine sample was assayed for albumin, protein, and creatinine to calculate (micro)albuminuria corrected for creatinine output (g/mol creatinine) and total protein level (g/24 h).

#### 2.1.1. Echocardiographic Measurements

The cardiac structure was assessed using a phased-array echocardiographic Doppler system (Hewlett-Packard Sonos 2000 and 2500; Hewlett-Packard Company, Palo Alto, CA, USA). A 2-dimensional, M-mode, and Doppler echocardiography was performed according to the guidelines of the American Society of Echocardiography [10]. Data were analyzed offline with the use of specific software (Excelera, Philips, The Netherlands). The left ventricular end-diastolic (LVEDd), and end-systolic diameters (LVESd), and the end-diastolic thickness of the interventricular septum (IVST), and the posterior wall (PWT) were all measured using the M-mode in parasternal long axis view (mm). To estimate the LVMi (g), the Devereux formula (0.8 × (1.04((LVEDd + PWT + IVST)^3^(LVEDd)^3^)) + 0.6) was used and indexed for BSA [11]. The relative wall thickness (RWT) was calculated as RWT = ((IVST + PWT)/LVEDd). The cut-off values of the RWT and LVMi were determined after calculating quintiles of the total formerly preeclamptic population. The highest, fifth quintile of RWT (=0.36) and LVMi (=84 g/m^2^) were used to define the cardiac geometry. Subjects with a RWT ≤ 0.36 or LVMi ≤ 84 g/m^2^ were defined as having healthy cardiac geometry. Individuals with RWT > 0.36 were considered as having relative concentric remodeling and subjects with LVMi > 84 g/m^2^ as having left ventricular hypertrophy. When having concomitant RWT > 0.36, we classified these individuals as concentric LVH, whereas RWT ≤ 0.36 as eccentric LVH. 

#### 2.1.2. Measurement of Circulating Markers 

C-reactive protein (CRP) was measured using a particle enhanced immunoturbidimetric assay. In this assay, the human CRP in the serum agglutinated with latex particles, which were coated with monoclonal anti-CRP antibodies from mice. The aggregates were determined turbidimetrically. From 2003 onwards, high sensitive C-reactive protein (hs-CRP) was also assessed by a multiarray detection system based on electro-chemiluminescence technology (MesoScaleDiscovery, SECTOR Imager 2400, Gaithersburg, MD, USA). Fibrinogen was measured based on the principle that the enzyme thrombin leads to the conversion of the soluble plasma protein fibrinogen into the insoluble polymer fibrin. The time the diluted plasma needed for clotting was inversely proportional to the fibrinogen concentration of the sample. The procedure was based on measuring the clotting time of the diluted plasma sample, after adding the enzyme thrombin. Clotting time was compared to the clotting time of a standardized fibrinogen preparation. Uric acid was measured by an enzymatic colorimetric test in which uric acid was cleaved into allantion and hydrogen peroxide by the enzyme uricase. Next, 4-aminophenazone was oxidized by hydrogen peroxide leading to the formation of a quinone-diimine dye due to the presence of peroxidase. The intensity of the color of the formed quinone-diimine dye was directly proportional to the concentration of uric acid in the sample. This intensity was measured by quantifying the increase in absorbance.

Metabolic syndrome was diagnosed based on the World Health Organization criteria as follows: the presence of hyperinsulinemia (fasting insulin ≥ 9.2 milliunits/L, fasting blood glucose ≥ 6.1 mmol/L, or HOMA-ir ≥ 2.2, or along with two or more of the following: (1) BMI ≥ 30 kg/m^2^, (2) dyslipidemia (triglycerides ≥ 1.69 mmol/L, or HDL ≤ 0.9 mmol/L), (3) hypertension: systolic blood pressure ≥ 140 mm Hg, or diastolic blood pressure ≥ 85 mm Hg, and (4) microalbuminuria (≥2.5 g/mol creatinine) or proteinuria (≥0.30 g per 24 h)) [12]. 

### 2.2. Statistical Analysis

Baseline characteristics were described as mean (standard deviation) and absolute number (percentage). Incomplete markers were imputed (17% of the total data) with stochastic regression imputation to prevent a decrease in statistical power and to decrease the likelihood of biased results associated with using only complete cases. We tested for differences between normal cardiac remodeling, concentric remodeling, eccentric left ventricle hypertrophy, concentric left ventricle hypertrophy using the analysis of variance (ANOVA) test for continuous variables or with the X^2^ and Fisher’s exact test for categorical variables. Because of the right-skewed distribution, the markers were logarithmically transformed because of severe skewness. The markers were described before transformation using the median (interquartile range). After transformation, levels of the markers were compared between the concentric remodeling, eccentric LVH, and concentric LVH groups to the cardiac unaffected healthy formerly preeclamptic group using analysis of covariance (ANCOVA), corrected for time interval. We considered a *p* value below 0.05 to be statistically significant. All computations were performed using SPSS Statistics version 21.0 (IBM Corp., Armonk, NY, USA). 

#### 2.2.1. Model Development

The markers were chosen based on expert consensus and a literature search. The potential markers were based on their pathophysiological mechanism. For each of the candidate markers, univariable linear regression was used. A liberal *p*-value of 0.2 was used to consider the potential marker important to include in the backward stepwise linear regression analysis, as per prediction modelling guidelines. Through backward stepwise elimination, candidate markers were excluded from the multivariable model that were not statistically significant (*p* < 0.05), leaving a more parsimonious model with the most significant markers. Univariable and multivariable regression analyses were performed to explore the associations between on the one hand SBP, PV, CRP, fibrinogen and uric acid as independent variables, and on the other hand RWT and LVMi as dependent variables. There is clear consensus about the maximum number of markers that can be used for multivariable modeling. At least ten events should be collected for each potential predictor that is to be evaluated in the multivariable regression analysis that can be validly included in a model [13]. Taking the power of our study into account, a total number of five markers in our final model was considered as acceptable.

#### 2.2.2. Model Performance 

Model calibration was explored using a scatter plot to compare predicted values to observed values. R-squared was computed to estimate what proportion of the variance in cardiac remodeling could be predicted using the model. The root mean squared error was used to quantify the average difference between observed and predicted values.

## 3. Results

### 3.1. Baseline Characteristics 

In this cross-sectional study, 752 former preeclamptic women were included. Of these 752 women, 515 (68%) had a normal cardiac geometry (control), 88 (12%) concentric remodeling (CR), 136 (18%) eccentric LVH, and 13 (2%) concentric LVH (Figure 1). The characteristics of the patients and controls are shown in Table 1. Patients with concentric LVH had on average, higher weight, BMI, and higher occurrence of obesity. Moreover, prehypertension was also more prevalent in the concentric LVH group compared to the control group (39% vs. 29%, respectively; *p* = 0.031). The use of antihypertensive treatment was higher in the eccentric and concentric LVH group compared with the control group (19% and 39% vs. 10%, respectively; *p* = 0.008 and *p* = 0.009). The diagnosis hypertension, based on either BP or use of antihypertensive medication was higher compared to the healthy former PE control group (30% and 54% vs. 15%, respectively; *p* < 0.001 and *p* < 0.001) but not compared to the concentric remodeled group. Most women where primiparous (87%) had early onset PE (70%), and preterm PE (80%) and did not have recurrent PE (96%).

### 3.2. Markers of Aberrant Cardiac Remodeling in Former Preeclamptic Women

Univariable linear regression analysis with LVMi and RWT as outcomes is shown in Table 2. We selected five variables (SBP, PV, fibrinogen, CRP, and uric acid) that were statistically significant according to our liberal *p*-value of 0.2 in the univariable analysis for the multivariable linear regression. Table 3 displays the values of the marker variables in the different groups. SBP was significantly higher in the CR group (115 mm Hg, *p* < 0.05), the eccentric LVH group (118 mm Hg, *p* < 0.01), and the concentric LVH group (136 mm Hg, *p* < 0.01) compared to the control group (114 mm Hg). PV was significantly higher in the concentric LVH group and the eccentric LVH group compared to the control group (1373 and 1371 vs. 1333 mL/BSA; *p* < 0.05) but not compared to the CR group (1276 mL/BSA). However, uric acid and fibrinogen were significantly higher in the concentric LVH group compared to the control group (0.32 vs. 0.26 mmol/L and 3.40 vs. 2.80 g/L; *p* < 0.05, respectively). 

Table 4 and Table 5 show two univariable and multivariable linear regression models of the entire sample, with RWT and LVMi as dependent variables and SBP, PV, CRP, fibrinogen and uric acid as independent variables. Multivariable analysis showed that both SBP and PV were independently correlated with RWT and LVMi ((regression coefficient (RC) = 0.06, 95% CI 0.03–0.08 and RC = 26.24, 95% CI = 18.07–34.41, *p* < 0.01, respectively) and (β = 0.02 95% CI = 0.04–0.00, *p* < 0.05 and β = 9.01, 95% CI = 2.81–15.35, *p* < 0.01, respectively)). CRP (β = 0.00, 95% CI = 0.00–0.01, *p*< 0.01) and uric acid (β = 0.02, 95% CI = 0.00–0.03, *p* < 0.05) were only independently related to RWT. Fibrinogen did not relate significantly to LVMi or RWT after correcting for other markers. 

To derive a multimarker model, we used the regression coefficients to formulate formulas for our population:RWT = 0.248 + 0.055 SBP (mm Hg) − 0.023 PV (mL/BSA) + 0.004 CRP (mg/L) + 0.003 Fibrinogen (g/L) + 0.017 Uric acid (mmol/L)
LVMi = −126.76 + 26.242 SBP (mm Hg) + 9.079 PV (mL/BSA) + 0.587 CRP (mg/L) + 3.620 Fibrinogen (g/L) − 4.55 Uric acid (mmol/L)

## 4. Discussion

This multimarker model involving both phenotypic characteristics and circulating biomarkers shows a potential role for SBP, PV, CRP, fibrinogen and uric acid as biomarkers to estimate aberrant cardiac remodeling in former preeclamptic women. Multivariable analysis showed that both SBP and PV were independently correlated with RWT and LVMi. SBP was higher in patients with CR, eccentric LVH, and concentric LVH compared to the control women. PV was significantly higher in patients with concentric and eccentric LVH compared to controls. CRP and uric acid were independently related to only RWT. Uric acid and fibrinogen were significantly higher in the patients with concentric LVH compared to controls. Fibrinogen did not relate to either LVMi or RWT. 

There is growing evidence that CVD are generally progressive disorders, proceeding from asymptomatic to symptomatic stages, with one of the principal manifestations being the change in geometry and function of the LV. Emerging evidence shows that concentric remodeling compared with eccentric remodeling, has considerable prognostic value in the development of CVD, with an odds ratio of 4.07 [14]. Not only do volume and pressure load determine cardiac remodeling, but neurohormonal factors also play an important role. Strongly upregulated brain natriuretic peptide (BNP), chronic adrenergic drive, elevated circulating catecholamines and attenuated beta-adrenergic receptor signaling increase the amount of myocardial protein synthesis, exceeding the amount of protein degradation [15,16,17,18,19]. 

We showed previously that pressure overload after PE results aggravates the risk for HF stage B 4 times already at prehypertension levels, even before hypertension develops [7,20]. In this study, we showed an independent correlation between SBP on the one hand and RWT and LVMi on the other hand. On the other hand, there was a negative correlation between PV and RWT and a positive relation between PV and LVMi. Besides, the lowest plasma volume was found in the concentric remodeling group. 

Inflammation plays a critical role in atherosclerosis and the inflammatory marker CRP has been shown to have a strong predictive power for CV events [21]. Interesting relations have been demonstrated between hypertensive status and systemic inflammatory activation [22]. In our entire study population, CRP exhibited positive relations with RWT and LVMi. In the population of former preeclamptic women, we showed a significantly higher CRP level in the concentric LVH group compared to all other groups. More inflammatory markers have been associated with an increased cardiovascular risk, but CRP specifically has attracted attention, as it can be measured easily and reliably and has a long half-life time. Moreover, fibrinogen seems to be an important predictor of cardiovascular events due to its role in the coagulation cascade and in acute vascular syndromes [23]. In this study, we found positive relations between fibrinogen and RWT and LVMi. Fibrinogen has an OR of 1.40 for concentric remodeling and even an OR of 3.05 for concentric LVH. Considerable epidemiologic studies have demonstrated a relationship between an increased uric acid level and CVD in the general population as well as patients with hypertension [24,25,26,27,28,29,30]. In our study population of former preeclamptic women, uric acid exhibited a positive relation with RWT. Appendix A provides a scientific rationale for each marker in our model. 

PE can be classified in different subtypes based on onset of disease (early-onset PE and late-onset PE). The type of cardiovascular dysfunction differs between women with early- and late-onset preeclampsia. Besides obstetric characteristics, PE can also be classified based on hemodynamic profile. In fact. hypertensive women with low CO and/or increased SVR had higher risk of developing preeclampsia sooner. These factors were consistently associated with increased risk of earlier development of preeclampsia even after adjusting for maternal and pregnancy characteristics such as blood pressure, ethnicity, and gestational age at assessment and initial diagnosis. Both low CO and increased SVR are strongly related to early-onset PE on the one hand, and concentric remodeling on the other hand. This finding is also in line with the finding concentric remodeling and LV hypertrophy are more prevalent after early onset PE than late onset PE [31]. Since the high-risk population consists of young, seemingly healthy women, personalized screening programs may have major clinical benefits in detecting individual women at risk to provide personalized prevention.

### Strength and Limitations

Several shortcomings of this study need to be stressed. First, the use of antihypertensive treatment was higher in all groups in comparison to the control group. Second, a study limitation was the use of a cross-sectional design. As a result, no longitudinal analyses could be performed on the changes within individuals over time. To draw more consistent conclusions on the association between the SBP, PV, the markers, and the influence on cardiac geometry, and the development over time, a longitudinal study design would be necessary to internally validate this. Future longitudinal studies may yield information on the predictive performance of markers for future outcomes and may have an impact on treatment of women early on after complicated pregnancy. Additionally, because some groups were relatively small compared to others, stark differences in statistical power between different tests may have complicated interpretation of the results. However, this is an unbiased sample from the records of our hospital and thus reflective of the proportion of individuals in each group. Testing and validation of those possibly effective factors is needed to identify the diagnostic and therapeutic capabilities. Furthermore, we do not have information about lactation time or physical activity of all women included in our study, which are known to affect the risk for CVD [32]. 

Future studies are needed to validate the predictive and therapeutic value of this model prospectively.

## 5. Conclusions

In summary, our study shows a correlation between elevated blood pressure, low plasma volume, CRP, uric acid, and fibrinogen as markers of aberrant cardiac remodeling in former preeclamptic women, suggesting a combination of mechanical (pressure and volume load) and biochemical (inflammation and oxidative stress) factors to be involved in worrisome chamber remodeling before clinical symptoms arise.

## Figures and Tables

**Figure 1 jcm-11-01900-f001:**
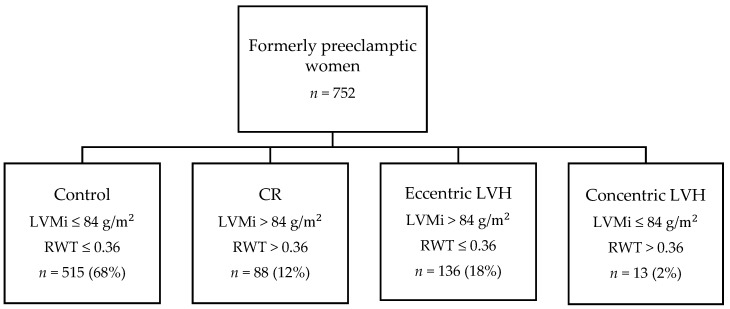
Flowchart of the study population. CR = concentric remodelling; LVH, left ventricular hypertrophy; LVMi, left ventricular mass index; RWT, relative wall thickness.

**Table 1 jcm-11-01900-t001:** Data are given as mean values (±SD) or *n*/*N* (%) of 752 formerly preeclamptic women with normal geometry (control), CR, eccentric LVH, and concentric LVH varying 6 months to 5 years postpartum. Original data file was used, and groups were compared to control group with the ANOVA test. Binary variables were tested with the X^2^ and Fisher’s exact test.

	Control (*n* = 515)	CR (*n* = 88)	*p*-Value	Eccentric LVH (*n* = 136)	*p*-Value	Concentric LVH (*n* = 13)	*p*-Value
Patient characteristics							
Age (years)	31 ± 4	32 ± 4	0.441	32 ± 4	0.049	32 ± 4	0.793
Postpartum (months)	16 ± 14	16 ± 12	0.636	16 ± 13	0.670	16 ± 11	0.996
GA at birth (weeks)	33 ± 4	33 ± 4	0.870	34 ± 4	0.569	31 ± 4	0.115
Weight (kg)	72 ± 14	72 ± 19	0.798	71 ± 13	0.386	85 ± 17	0.001
BMI (kg/m^2^)	25 ± 5	26 ± 6	0.253	25 ± 4	0.964	30 ± 5	<0.001
Obesity, BMI ≥ 30 kg/m^2^, *n* (%)	77/515 (15)	20/88 (23)	0.067	14/136 (10)	0.164	6/13 (46)	0.009
Smoking, *n* (%)	10/123 (8)	2/30 (7)	0.570	2/58 (3)	0.343	10/13 (77)	1.000
Alcohol, *n* (%)	44/162 (27)	13/38 (34)	0.386	39/89 (44)	0.007	6/13 (46)	0.192
Family history of CVD, *n* (%)	226/503 (45)	47/85 (55)	0.076	71/135 (53)	0.113	6/13 (46)	0.930
Prehypertension, *n* (%)	145/509 (29)	24/88 (27)	0.975	37/131 (28)	0.284	5/13 (39)	0.031
Antihypertensive treatment, *n* (%)	53/513 (10)	15/88 (17)	0.066	25/133 (19)	0.008	5/13 (39)	0.009
Hypertension, *n* (%)	29/509 (6)	9/88 (10)	0.114	27/133 (21)	<0.001	6/13 (46)	<0.001
Hypertension based on BP or AT, *n* (%)	76/513 (15)	19/88 (22)	0.107	40/133 (30)	<0.001	7/13 (54)	<0.001
Blood pressure and metabolic variables							
SBP (mm Hg)	115 ± 12	118 ± 16	0.026	121 ± 15	<0.001	136 ± 15	<0.001
DBP (mm Hg)	71 ± 9	75 ± 10	0.091	74 ± 11	<0.001	82 ± 8	0.874
MAP (mm Hg)	87 ± 10	91 ± 11	0.046	92 ± 12	<0.001	101 ± 11	0.120
HR (bpm)	73 ± 25	75 ± 12	0.568	70 ± 10	0.465	78 ± 11	0.881
TC mmol*L^−1^	4.8 ± 0.9	4.9 ± 1.0	0.146	4.9 ± 0.9	0.326	5.6 ± 1.3	0.198
HDL mmol*L^−1^	1.3 ± 0.3	1.3 ± 0.4	0.719	1.4 ± 0.4	0.870	1.2 ± 0.3	0.931
LDL mmol*L^−1^	2.9 0.8	3.0 ± 0.9	0.639	3.1 ± 0.9	0.151	3.7 ± 1.4	0.071
TG mmol*L^−1^	1.0 ± 0.8	1.1 ± 0.7	0.488	0.9 ± 0.5	0.015	1.8 ± 1.4	<0.001
Glucose mU*L^−1^	5.1 ± 0.7	5.4 ± 1.7	<0.001	5.2 ±1.1	0.136	5.6 ± 1.0	0.003
Insulin mU*L^−1^	10 ± 7.3	13 ± 19	<0.001	9.8 ± 5.3	0.074	17 ± 12	0.001
HbA1c mmol*L^−1^	5.2 ± 0.5	5.4 ± 0.7	0.007	5.4 ± 0.6	0.857	5.4 ± 0.6	0.136
PV index (mL/m^−2^)	1333 ± 173	1276 ± 162	0.006	1371 ± 200	0.026	1373 ± 155	0.408
Obstetric variables							
Birth weight (g)	1936 ± 833	1907 ± 925	0.767	1886 ± 831	0.541	1690 ± 1058	0.298
Primiparous *n* (%)	458/515 (89)	73/88 (83)	0.049	112/136 (83)	0.085	11/13 (85)	0.050
Early onset PE *n* (%)	349/515 (68)	63/88 (72)	0.476	98/136 (72)	0.337	10/13 (77)	0.485
Recurrence PE	18/515 (4)	4/88 (5)	0.640	8/136 (6)	0.201	1/13 (8)	0.428
Preterm PE	415/515 (81)	69/88 (78)	0.636	110/136 (80)	0.937	11/13 (85)	0.716
SGA neonate, *n* (%)	71/317 (22)	19/65 (29)	0.237	31/131 (24)	0.771	2/11 (18)	1.000

AT, antihypertensive treatment; BMI, body mass index; BP, blood pressure; CR, concentric remodeling; DBP, diastolic blood pressure; GA, gestational age; HDL, high-density lipoprotein; HR, heart rate; LDL, low-density lipoprotein; LVH, left ventricular hypertrophy; MAP, mean arterial pressure; PE, preeclampsia; PV, plasma volume indexed for body surface area (BSA); SBP, systolic blood pressure; SGA, small-for-gestational age; TC, total cholesterol; TG, triglycerides.

**Table 2 jcm-11-01900-t002:** Univariable linear regression analysis with LVMi and RWT as outcome using transformed data set.

	Univariate Analysis LVMi		Univariate Analysis for RWT	
	B (CI)	*p*-Value	B (CI)	*p*-Value
ATIII (%)	0.003 (−0.132–0.138)	0.963	0.001 (0.000–0.003)	0.127
DBP (mm Hg)	0.007 (0.001–4.654)	0.317	0.823 (−0.512–2.342)	0.456
SBP (mm Hg)	0.216 (−0.076–0.286)	0.003	0.001 (0.000–0.006)	0.022
PV (mL/BSA)	0.105 (−0.076–0.286)	0.054	0.001 (−0.001–0.002)	0.032
Estrogen (nmol/L)	−5.654 (−7.874–1.565)	0.424	0.007 (−0.022–0.036)	0.618
CRP (mg/L)	0.108 (−0.433–0.650)	0.158	0.002 (0.000–0.004)	0.068
Homocysteine basal (umol/L)	−0.300 (−0.684–0.084)	0.625	0.000 (−0.002–0.001)	0.365
Fibrinogen (g/L)	0.469 (−2.678–3.616)	0.169	−0.003 (−0.016–0.010)	0.041
Progesterone (nmol/L)	0.075 (−0.064–0.214)	0.287	0.000 (−0.001–0.000)	0.628
Uric acid (mmol/L)	9.799 (−8.730–12.328)	0.053	0.082 (−0.032–0.197)	0.058

ATIII, Antithrombine III; CRP, C-reactive protein; DBP, diastolic blood pressure; LVMi, left ventricular mass index; PV, plasma volume indexed for BSA; RWT, relative wall thickness; SBP, systolic blood pressure.

**Table 3 jcm-11-01900-t003:** Data are given as median (interquartile range) of the marker variables in 752 formerly preeclamptic women with normal geometry (control), eccentric LVH, and concentric LVH 6 months to 5 years postpartum. Untransformed marker data were used to calculate median (interquartile range). Transformed marker data were used to compare groups to control using ANCOVA.

	Control (*n* = 511)	CR (*n* = 83)	Eccentric LVH (*n* = 134)	Concentric LVH (*n* = 13)
PV (mL/BSA)	1237 (1234–1442)	1284 (1190–1372)	1362 (1283–1464)	1396 (1240–1497)
SBP (mm Hg)	114 (107–122)	115 (109–125)	118 (108–132)	136 (126–148)
Uric acid (mmol/L)	0.26 (0.23–0.31)	0.27 (0.25–0.31)	0.24 (0.22–0.29)	0.32 (0.27–0.36)
CRP (mg/L)	1.02 (0.50–2.63)	1.46 (0.50–3.76)	1.18 (0.50–2.38)	4.10 (2.08–4.10)
Fibrinogen (g/L)	2.80 (2.50–3.30)	3.10 (2.60–3.40)	2.90 (2.60–3.40)	3.40 (3.05–3.75)

CR, concentric remodeling; LVH, left ventricular hypertrophy; PV, plasma volume indexed for BSA; SBP, systolic blood pressure.

**Table 4 jcm-11-01900-t004:** Univariable and multivariable linear regression model with RWT as outcome using transformed data set. The regression coefficients were used to formulate a formula: RWT = 0.248 + 0.055 SBP (mm Hg) − 0.023 PV (mL/BSA) + 0.004 CRP (mg/L) + 0.003 Fibrinogen (g/L) + 0.017 Uric acid (mmol/L).

	Univariable		Multivariable *	
	B (CI)	*p*-Value	B (CI)	*p*-Value
Intercept	-	-	0.248	-
SBP (mm Hg)	0.069 (0.042–0.095)	<0.01	0.055 (0.029–0.082)	<0.01
PV (mL/BSA)	−0.029 (−0.050–−0.008)	<0.01	−0.023 (−0.044–0.003)	<0.05
CRP (mg/L)	0.007 (0.004–0.009)	<0.01	0.004 (0.001–0.008)	<0.01
Fibrinogen (g/L)	0.026 (0.013–0.040)	<0.01	0.003 (−0.013–0.020)	0.72
Uric acid (mmol/L)	0.027 (0.013–0.042)	<0.01	0.017 (0.002–0.032)	<0.05

* $R^{2}$ = 0.071, the R-square gives the amount of variability in the outcome that is accounted for by the predicted variables. B, beta coefficient; CI, confidence interval; SBP, systolic blood pressure; PV, plasma volume indexed for body surface area (BSA).

**Table 5 jcm-11-01900-t005:** Univariable and multivariable linear regression model with LVMi as outcome using transformed data set. The regression coefficients were used to formulate a formula: LVMi = −126.76 + 26.242 SBP (mm Hg) + 9.079 PV (mL/BSA) + 0.587 CRP (mg/L) + 3.620 Fibrinogen (g/L) − 4.55 Uric acid (mmol/L).

	Univariable		Multivariable *	
	B (CI)	*p*-Value	B (CI)	*p*-Value
Intercept	-	-	−126.755	-
SBP (mm Hg)	26.824 (18.751–34.897)	<0.01	26.242 (18.072–34.412)	<0.01
PV (mL/BSA)	6.346 (−0.068–12.760)	0.05	9.079 (2.809–15.349)	<0.01
CRP (mg/L)	1.105 (0.314–1.896)	<0.01	0.587 (−0.389–1.562)	0.24
Fibrinogen (g/L)	5.883 (1.706–10.060)	<0.01	3.620 (−1.445–8.684)	0.16
Uric acid (mmol/L)	−1.288 (−5.894–3.318)	0.58	−4.55 (−9.23–0.13)	0.06

* $R^{2}$ = 0.073, the R-square gives the amount of variability in the outcome that is accounted for by the marker variables. B, beta coefficient; CI, confidence interval; SBP, systolic blood pressure; PV, plasma volume indexed for body surface area (BSA).

## Data Availability

All data generated or analyzed during the current study are included in this manuscript.

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
