# Peer review of "A Multimarker Model for Aberrant Cardiac Geometry after Preeclampsia"

_jcm, 2022, doi:10.3390/jcm11071900_

Round 1
Reviewer 1 Report
Dear Authors,
This study targets an important research topic and includes a large study population size. However, there are some limitations which may affect the clinical interpretation of the results. I recommend the following suggestions. Thank you for considering my feedback.
- Methods:
- Study population: please add some details about the participants:
- Did the authors include only women with history of preeclampsia without history of chronic hypertension? Please specify. Since chronic hypertension may be underdiagnosed especially in case of late prenatal care access, please specify which methods were used to exclude those women with history of superimposed preeclampsia in a context of chronic hypertension in pregnancy.
- Please specify the exclusion criteria. Did the authors exclude women with certain diseases at enrollment? Such as severe hypertension, respiratory diseases…?
- Study method:
- Study population: please add some details about the participants:
Since the study included participants who had preeclampsia between 1996-2014, the diagnostic criteria to define preeclampsia are based on relatively “old” criteria (reference 12, publication year 2001) which are slightly different compared to the most recent criteria (Gestational Hypertension and Preeclampsia: ACOG Practice Bulletin, Number 222. Obstet Gynecol. 2020 Jun;135(6):e237-e260). The authors may consider mentioning about this in the discussion. The reason is that changing in diagnostic criteria over the years (even minor changes) may affect reproducibility of the study/validation of the results.
- Model development:
- Please consider changing the term “predictors” with markers, or another term. The reason is that this study is cross-sectional, not longitudinal, so it is hard to establish whether PV, CPR…predict cardiac remodeling, since all the parameters are assessed at the same time.
- Please consider adding a brief scientific rationale of each selected marker to be tested by the model. It can be a simple table (supplemental material eventually).
- Results:
- Please specify whether the study groups (control, CR…) differ in regard to:
- Parity: the rationale is that multigravida women have an increased risk for cardiovascular diseases.
- Number of previous pregnancy complicated by preeclampsia. Did all the women included in the study have history of one pregnancy complicated by preeclampsia, or one or more pregnancies complicated by PE?
- Physical activity
- Lactation time: the rationale is that women with longer lactation time have a lower risk for cardiovascular diseases after pregnancies, compared to women with shorter lactation time and women whit no history of breastfeeding (Klingberg S, Brekke HK, Winkvist A, et al. Parity, weight change, and maternal risk of cardiovascular events. Am J Obstet Gynecol 2017;216: 172.e1-15.). (Wiklund P, Xu L, Lyytikäinen A, Saltevo J, Wang Q, Völgyi E, Munukka E, Cheng S, Alen M, Keinänen-Kiukaanniemi S, Cheng S. Prolonged breast-feeding protects mothers from later-life obesity and related cardio-metabolic disorders. Public Health Nutr. 2012 Jan;15(1):67-74).
- Please specify whether the study groups (control, CR…) differ in regard to:
If the above information is not available, please consider discussing about this in the limitation.
- Table 1: please clarify postpartum year. Are the number mean 16 year and SD 14 correct? The study method states that data were collected between six months and five year postpartum.
- Figure 1-note: does LVH refer to left ventricular hypotrophy? Or hypertrophy? Please also check the notes of the tables.
Discussion: Since there is now evidence that there are different forms of preeclampsia, based not just on the gestational age at onset, but also on associated cardiovascular parameters (with normal or lower cardiac output, Kalafat E, Perry H, Bowe S, Thilaganathan B, Khalil A. Prognostic Value of Maternal Cardiovascular Hemodynamics in Women With Gestational Hypertension and Chronic Hypertension in Pregnancy. Hypertension. 2020 Aug;76(2):506-513), can the authors discuss about how different types of preeclampsia may affect the cardiovascular remodeling in the postpartum? Do the authors have data about maternal echocardiogram performed in pregnancy of the study population? If so, it would be very important to add, as secondary analysis, an association between maternal echocardiogram in pregnancy and echocardiogram done in the postpartum.
Author Response
Q1. This study targets an important research topic and includes a large study population size. However, there are some limitations which may affect the clinical interpretation of the results. I recommend the following suggestions. Thank you for considering my feedback. Methods: Study population: please add some details about the participants: Did the authors include only women with history of preeclampsia without history of chronic hypertension? Please specify. Since chronic hypertension may be underdiagnosed especially in case of late prenatal care access, please specify which methods were used to exclude those women with history of superimposed preeclampsia in a context of chronic hypertension in pregnancy.
A1. We thank the reviewer for this relevant suggestion. We indeed only included women with a history of preeclampsia without a history of chronic hypertension. During our cardiovascular assessment we specifically ask about medical history. Those reporting chronic hypertension where excluded. Since most of the included women had their delivery and antenatal care in many different hospitals in the Netherlands or with a midwife, we do not have a clear overview on whom of the women developed high blood pressure before 20 weeks of gestation.
Q2. Please specify the exclusion criteria. Did the authors exclude women with certain diseases at enrollment? Such as severe hypertension, respiratory diseases…?
A2. Indeed, we excluded women with certain pre-existing relevant diseases. We added the exclusion criteria on page 2 lines 89-90 in the methods section as follows: “Exclusion criteria were chronic hypertension, auto-immune disease, or kidney disease prior to the first pregnancy.”
Q3. Study method: Since the study included participants who had preeclampsia between 1996-2014, the diagnostic criteria to define preeclampsia are based on relatively “old” criteria (reference 12, publication year 2001) which are slightly different compared to the most recent criteria (Gestational Hypertension and Preeclampsia: ACOG Practice Bulletin, Number 222. Obstet Gynecol. 2020 Jun;135(6):e237-e260). The authors may consider mentioning about this in the discussion. The reason is that changing in diagnostic criteria over the years (even minor changes) may affect reproducibility of the study/validation of the results.
A3. We thank the reviewer for this relevant comment. Since we included women with preeclampsia between 1996 and 2014, we used diagnostic criteria to define preeclampsia that are not based on the most recent guidelines. Therefore, no discrepancy was introduced in the definition used before and after 2020. Moreover, in clinical practice in the Netherlands, still the classic definition is most widely used.
Q4. Model development: Please consider changing the term “predictors” with markers, or another term. The reason is that this study is cross-sectional, not longitudinal, so it is hard to establish whether PV, CPR…predict cardiac remodeling, since all the parameters are assessed at the same time.
A4. We thank the reviewer for this relevant comment. We edited the suggestions in the revised manuscript and replaced the term ‘predictor’ to the term marker.
Q5. Please consider adding a brief scientific rationale of each selected marker to be tested by the model. It can be a simple table (supplemental material eventually).
A5. We thank the reviewer for this relevant comment. In response to the reviewer’s comment, we added Table S1 in which a scientific explanation is given for each marker we have selected for our model with references included. We referred in the main manuscript to table S1 on page 10, line 478-479 as follows: “Table S1 (in supplemental material section) provides a scientific rationale for each marker of our model.
Q6. Results: Please specify whether the study groups (control, CR…) differ in regard to:
- Parity: the rationale is that multigravida women have an increased risk for cardiovascular diseases.
- Number of previous pregnancy complicated by preeclampsia. Did all the women included in the study have history of one pregnancy complicated by preeclampsia, or one or more pregnancies complicated by PE?
- Physical activity
- Lactation time: the rationale is that women with longer lactation time have a lower risk for cardiovascular diseases after pregnancies, compared to women with shorter lactation time and women whit no history of breastfeeding (Klingberg S, Brekke HK, Winkvist A, et al. Parity, weight change, and maternal risk of cardiovascular events. Am J Obstet Gynecol 2017;216: 172.e1-15.). (Wiklund P, Xu L, Lyytikäinen A, Saltevo J, Wang Q, Völgyi E, Munukka E, Cheng S, Alen M, Keinänen-Kiukaanniemi S, Cheng S. Prolonged breast-feeding protects mothers from later-life obesity and related cardio-metabolic disorders. Public Health Nutr. 2012 Jan;15(1):67-74).
If the above information is not available, please consider discussing about this in the limitation.
A6. We thank the reviewer for this relevant remark. We added the variables “primiparous, recurrence PE and physical activity” in the baseline characteristics Table 1 and in the results section on page 5, line 294-296 as follows: “Most women where primiparous (87%) had early onset PE (70%), and preterm PE (80%)and did not have a recurrent PE (96%)”.
Unfortunately, we do not have information about physical activity and lactation time of all women included in our study. We classified this as a limitation of our study on page 9 line 507-509 as follows: Also, we do not have information about lactation time or physical activity of all women included in our study, which are known to affect the risk for CVD.
Q7. Table 1: please clarify postpartum year. Are the number mean 16 year and SD 14 correct? The study method states that data were collected between six months and five years postpartum.
A7. We thank the reviewer for this relevant comment. The number mean 16 and SD 14 in Table 1 are correct. However, it should be month instead of year. Therefore, we changed year to month in the table. Q8. Figure 1-note: does LVH refer to left ventricular hypotrophy? Or hypertrophy? Please also check the notes of the tables.
A8. We thank the reviewer for this relevant comment. LVH refers to left ventricular hypertrophy instead of hypotrophy. We have specified the abbreviation in the introduction and changed hypotrophy to left ventricular hypertrophy in the notes of the tables.
Q9. Discussion: Since there is now evidence that there are different forms of preeclampsia, based not just on the gestational age at onset, but also on associated cardiovascular parameters (with normal or lower cardiac output, Kalafat E, Perry H, Bowe S, Thilaganathan B, Khalil A. Prognostic Value of Maternal Cardiovascular Hemodynamics in Women With Gestational Hypertension and Chronic Hypertension in Pregnancy. Hypertension. 2020 Aug;76(2):506-513), can the authors discuss about how different types of preeclampsia may affect the cardiovascular remodeling in the postpartum?
A9. We thank the reviewer for this relevant comment. In response to the reviewer’s comment, we added some lines on the different forms of preeclampsia in the discussion section on page 9, line 478-488 of the revised manuscript as follows: “PE can be classified in different subtypes based on onset of disease (early-onset PE and late-onset PE). The type of cardiovascular dysfunction differs between women with early- and late-onset preeclampsia. Besides obstetric characteristics, PE can also be classified based on hemodynamic profile. In fact. hypertensive women with low CO and/or increased SVR had higher risk of developing preeclampsia sooner. These factors were consistently associated with increased risk of earlier development of preeclampsia even after adjusting for maternal and pregnancy characteristics such as blood pressure, ethnicity, and gestational age at assessment and initial diagnosis. As both low CO and increased SVR are strongly related to concentric remodeling which is in line with the finding that after early-onset PE concentric remodeling and LV hypertrophy are more prevalent than late onset PE (Kalafat, E., et al., Prognostic Value of Maternal Cardiovascular Hemodynamics in Women With Gestational Hypertension and Chronic Hypertension in Pregnancy. Hypertension, 2020. 76(2): p. 506-513).
Q10. Do the authors have data about maternal echocardiogram performed in pregnancy of the study population? If so, it would be very important to add, as secondary analysis, an association between maternal echocardiogram in pregnancy and echocardiogram done in the postpartum.
A10. We thank the reviewer for this relevant comment. Unfortunately, we do not have any additional maternal echocardiogram performed in the pregnancy of our study population.
Reviewer 2 Report
This is an interesting paper about the prediction of aberrant cardiac geometry after preeclampsia. For this, the authors underwent a cross-sectional study that included 752 women with a history of preeclampsia. These participants had a cardiovascular evaluation between six months and five years postpartum, assessing systolic and diastolic blood pressure, plasma volume, uric acid, C-reactive protein and fibrinogen. With all these information, the objective of the study was to develop a formula to estimate the aberrant cardiac remodeling. Nevertheless, I have several questions for the authors:
- The authors have consider just preeclampsia itself as a risk factor, although most probably the majority of these women had already other risk factors for cardiovascular disease beyond pregnancy. It would have been also interesting to have information about the control of blood pressure in the first 6 months after birth. If these women were recruited at the six weeks postpartum follow up, is there any information regarding the blood pressure of these women?
- On the other hand, it seems that even the maternal characteristics described in Table 1 are mixed in time in time, for example, BMI is at 6 months or 5 years? This should be addressed in the text.
- The authors describe different models for the prediction of aberrant cardiac remodeling. However, it would be more than interesting the detection rate.
- It would have been also very interesting to compare these women with those with normal pregnancy. Not only that, but also any differences between early or late preeclampsia, that by the way the authors include these definitions in the material and also probably in preterm versus late preeclampsia. Also, how many cases of all these were included?
Author Response
Q1. This is an interesting paper about the prediction of aberrant cardiac geometry after preeclampsia. For this, the authors underwent a cross-sectional study that included 752 women with a history of preeclampsia. These participants had a cardiovascular evaluation between six months and five years postpartum, assessing systolic and diastolic blood pressure, plasma volume, uric acid, C-reactive protein and fibrinogen. With all these information, the objective of the study was to develop a formula to estimate the aberrant cardiac remodeling. Nevertheless, I have several questions for the authors:
The authors have consider just preeclampsia itself as a risk factor, although most probably the majority of these women had already other risk factors for cardiovascular disease beyond pregnancy. It would have been also interesting to have information about the control of blood pressure in the first 6 months after birth. If these women were recruited at the six weeks postpartum follow up, is there any information regarding the blood pressure of these women?
A1. We thank the reviewer for this compliment and the suggestions for improvement. Unfortunately, despite the very interesting suggestion, we do not have any additional information about blood pressure in the first 6 months after birth as most of these women did their regular postpartum clinical follow up in very different hospitals spread over the Netherlands.
Q2. On the other hand, it seems that even the maternal characteristics described in Table 1 are mixed in time, for example, BMI is at 6 months or 5 years? This should be addressed in the text.
A2. Each women had the complete cardiovascular assessment on one session which means that all variables are retrieved from the same postpartum interval in each women. Given the cross-sectional design, not all women were included exactly at 6 months postpartum. The moment of inclusion ranges from six months to five years postpartum. In response to the reviewer’s comment, we edited this matter in Table 1 in our revised manuscript.
Q3. The authors describe different models for the prediction of aberrant cardiac remodeling. However, it would be more than interesting the detection rate.
A3. Unfortunately, we do not completely understand the comment of the reviewer. If the reviewer means to validate the model for predictive and therapeutic purpose, we must say that this is indeed a valuable step and is subject for upcoming studies. For now it goes beyond the scope of this paper as we do not have the suitable study design yet.
Q4. It would have been also very interesting to compare these women with those with normal pregnancy. Not only that, but also any differences between early or late preeclampsia, that by the way the authors include these definitions in the material and also probably in preterm versus late preeclampsia. Also, how many cases of all these were included?
A4. We thank the reviewer for this relevant remark. For this study, 752 women with a history of PE were included to assess the association between aberrant cardiac geometry after PE. In response to the reviewer’s comment, we added the variables early onset PE and preterm PE in Table 1 and our results section on page 5, line 299-301 as follows: Most women where primiparous (87%) had early onset PE (70%), and preterm PE (80%) and did not have recurrent PE (96%).
Round 2
Reviewer 2 Report
Thank you for the corrections of the paper.
This manuscript is a resubmission of an earlier submission. The following is a list of the peer review reports and author responses from that submission.